# A systematic review and meta-analysis on the effectiveness of an invasive strategy compared to a conservative approach in patients > 65 years old with non-ST elevation acute coronary syndrome

**Joan Dymphna P. Reaño[1]\***, **Louie Alfred B. Shiu[1]**, **Karen V. Miralles[1]**, **Maria Grethel C. Dimalala[2]**, **Noemi S. Pestaño[1]**, **Felix Eduardo R. Punzalan[1,3]**, **Bernadette Tumanan-Mendoza[1]**, **Michael Joseph T. Reyes[1,2]**, **Rafael R. Castillo[1,4,5]**

1 Adult Cardiology, Manila Doctors Hospital, Manila, Philippines, 2 Interventional Cardiology, Manila Doctors Hospital, Manila, Philippines, 3 Division of Cardiology, Department of Medicine, Philippine General Hospital, College of Medicine University of the Philippines, Manila, Philippines, 4 Cardiovascular Medicine, Adventist University of the Philippines, Silang, Philippines, 5 FAME Leaders Academy, Makati, Philippines

\* jdp.reano@gmail.com, medicalfiles.inquirer@gmail.com

**Data Availability Statement:** All relevant data are within the manuscript and its Supporting Information files.

## Abstract

### Background

Patients 65 years old and older largely represent (>50%) hospital-admitted patients with acute coronary syndrome (ACS). Data are conflicting comparing efficacy of early routine invasive (within 48–72 hours of initial evaluation) versus conservative management of ACS in this population.

### Objective

We aimed to determine the effectiveness of routine early invasive strategy compared to conservative treatment in reducing major adverse cardiovascular events in patients 65 years old and older with non-ST elevation (NSTE) ACS.

### Data sources

We conducted a systematic review of randomized controlled trials (RCTs) through PubMed, Cochrane, and Google Scholar database.

### Study selection

The studies included were RCTs that evaluated the effectiveness of invasive strategy compared to conservative treatment among patients ≥ 65 years old diagnosed with NSTEACS. Studies were included if they assessed any of the following outcomes of death, cardiovascular mortality, myocardial infarction (MI), stroke, recurrent angina, and need for revascularization. Six articles were subsequently included in the meta-analysis.

**Funding:** The author(s) received no specific funding for this work.

**Competing interests:** I have read the journal's policy and the authors of this manuscript have the following competing interests: RRC is a member of advisory board or speakers' pool of Servier, Boehringer Ingelheim, Menarini, LRI-Therapharma, Sanofi, UAP Pharma, Unilab; MTR is a member of speakers' pool of Novartis, Servier, Astra Zeneca; the rest declare no conflict of interest. This does not alter our adherence to PLOS ONE policies on sharing data and materials.

### Data extraction

Three independent reviewers extracted the data of interest from the articles using a standardized data collection form that included study quality indicators. Disparity in assessment was adjudicated by another reviewer.

### Data synthesis

All pooled analyses were initially done using Fixed Effects model. For pooled analyses with significant heterogeneity ($I^2 \geq 50\%$), the Random Effects model was used. A total of 3,768 patients were included, 1,986 in the invasive strategy group, and 1,782 in the conservative treatment group.

### Results

Meta-analysis showed less incidence of revascularization in the invasive (2%) over conservative treatment groups (8%), with overall risk ratio of 0.29 (95% CI 0.14 to 0.59). Across all pooled studies, no significant effect of invasive strategy on all-cause mortality, cardiovascular mortality, stroke, and MI was observed. Only one study assessed the outcome of recurrent angina.

### Conclusion

There was a significantly lower rate of revascularization in the invasive strategy group compared to the conservative treatment group. In the reduction of all-cause mortality, cardiovascular mortality, MI, and stroke there was no significant effect of invasive strategy versus conservative treatment. This finding does not support the bias against early routine invasive intervention in patients $\geq 65$ years old with NSTEACS. Further studies focusing on these patients with larger population sizes are still needed.

### Introduction

Based on the World Health Organization's Global Burden of Disease report, ischemic heart disease (IHD) is the overall leading cause of death worldwide [1]. Although the annual number of hospital discharges for acute coronary syndromes (ACS) in developed countries has declined slowly over the past two decades, the number has increased in developing countries [2]. In the Philippines, cardiovascular disease (CVD) remains the leading cause of mortality [3]. The Philippine Heart Association ACS registry reported that ACS is prevalent in the age range 51–70, with mean age group of 66 years old [3].

The most recent American College of Cardiology/American Heart Association (ACC/AHA 2014) and the European Society of Cardiology (ESC 2015) guidelines for non–ST segment elevation ACS (NSTEACS) reflect medical advancements in therapeutics and strategies of care leading to improved survival in ACS, but this was mainly observed in relatively younger individuals (<65 years of age) and in men. These guidelines emphasize intensive and early medical and interventional therapy, particularly for those at high risk [4–6].

The 2014 AHA/ACC NSTEACS Guidelines generally recommend that older patients with NSTEACS should be treated with goal-directed medical therapy, together with an early invasive strategy, and revascularization as appropriate [5]. The 2015 ESC Guidelines for the

Management of ACS, on the other hand, recommend that decisions on elderly patients with NSTEACS should be based on ischemic and bleeding risks, estimated life expectancy, comorbidities, quality of life, patient values and preferences, and the estimated risks and benefits of revascularization [6]. Despite the guidelines, older patients are less likely to undergo procedures after an NSTEACS than younger patients due in part to patient and practitioner concerns about the increased risk of complications [7–9].

Due to conflicting results of studies, lack of specific recommendations from the abovementioned guidelines, and the paucity of data on early invasive strategy versus conservative treatment for NSTEACS in patients ≥ 65 years old, this meta-analysis was conducted to focus on this special population to compare benefits and risks of early invasive therapy versus conservative management.

## Research question

Among elderly patients aged ≥ 65 years old with NSTEACS, how effective is invasive strategy compared to conservative treatment in preventing major adverse cardiovascular events (MACE)?

## Objectives

### General

To determine the effectiveness of invasive strategy compared to conservative treatment in reducing MACE among elderly patients with NSTEACS.

### Specific

Among patients ≥ 65 years old with NSTEACS, to determine the effectiveness of invasive strategy compared to conservative treatment, in 6 months (short-term) to 15 years (long-term), in reducing:

a. Death or all-cause mortality;

b. Cardiovascular mortality;

c. Myocardial infarction (MI);

d. Stroke;

e. Recurrent angina;

f. Need for revascularization.

## Methodology

### Study registration

Prior to the conduct of the research, the study was registered and approved by the Committee on Research (CORES) of Manila Doctors Hospital.

### Criteria for considering studies for this review

The studies included were RCTs that evaluated the effectiveness of invasive strategy compared to conservative treatment among patients ≥ 65 years old diagnosed with NSTEACS. Studies were included if any of the outcomes assessed were: death, cardiovascular mortality, MI, stroke, recurrent angina, and need for revascularization.

## Definition of terms

1. **Invasive strategy or early invasive strategy**–Routine early cardiac catheterization (within 48–72 hours of initial evaluation) followed by percutaneous coronary intervention (PCI) or coronary artery bypass graft (CABG) depending on the coronary anatomy within the patient's hospital admission for the index event on top of continuing medical therapy.

2. **Conservative treatment**—Initial optimal medical management, with cardiac catheterization reserved for patients with recurrent ischemia at rest or after a non-invasive stress test, followed by revascularization if the anatomy is suitable.

3. **Elderly patients**–Patients aged 65 years or older (WHO, 2000), with or without comorbidities.

4. **Non-ST elevation acute coronary syndrome (NSTEACS)**–A clinical manifestation of ischemic heart disease, which includes non-ST elevation MI and unstable angina, both presenting as: (1) sudden onset of symptoms at rest lasting at least 10 minutes; (2) severe pain, pressure, or discomfort in the chest; (3) accelerating pattern of chest pain that develops more frequently, more severe, or awakens patient from sleep.

5. **Non-ST elevation MI (NSTEMI)**–Patients with abovementioned symptoms without ST-segment elevation in at least 2 contiguous electrocardiogram (ECG) leads but with elevation of myocardial biomarkers > 99% percentile of normal

6. **Unstable angina**—Patients with abovementioned symptoms without ST-segment elevation in at least 2 contiguous electrocardiogram (ECG) leads but without elevation of myocardial biomarkers

## Search methods for identification of studies

Systematic computerized search (S1 Appendix) was performed using the Pubmed, Google Scholar, and Cochrane databases. MESH and free text of the following main key terms were used: "randomized controlled trials", "elderly", "non-ST elevation acute coronary syndrome", "invasive strategy", "conservative management", "invasive strategy versus conservative strategy", "major adverse cardiovascular events", "all-cause mortality", "cardiovascular mortality", "myocardial infarction", "stroke", "recurrent angina", "need for revascularization". The last search was done on September 1, 2019.

Eligibility assessment was performed independently in a standard manner by three reviewers. The literature search identified 151 relevant articles. Of the 151 articles, 34 were excluded due to different intervention since they did not involve comparing invasive versus conservative management in ACS. Among the 117 articles left, 30 articles were excluded due to different outcomes being assessed (i.e. arterial access site-related outcomes, risk for CABG, risk for heart failure, abnormal tissue perfusion). Among the 87 articles left, 51 articles subsequently were excluded due to different methods (i.e. observational studies, post-hoc analysis, cost-study, adherence study, registry data, outcome research study). Furthermore, among the 36 articles left, 23 were excluded due to different population (i.e. involved patients who have STEMI, post-MI, stable IHD, sepsis). Thirteen articles were then fully reviewed for eligibility. Among the 13 articles reviewed, 2 articles were possibly eligible but these are ongoing RCTs (MOSCA-FRAIL and DEAR-OLD Trials) with no published data yet at time of writing; hence, excluded [10,11]. Among the remaining 11 articles, 2 articles were excluded due to different intervention while 2 articles were excluded due to different population. One article was possibly eligible but did not report the event rates per treatment group (details for the titles of the studies and reasons for exclusion are listed in S2 Appendix). To access needed data in this particular

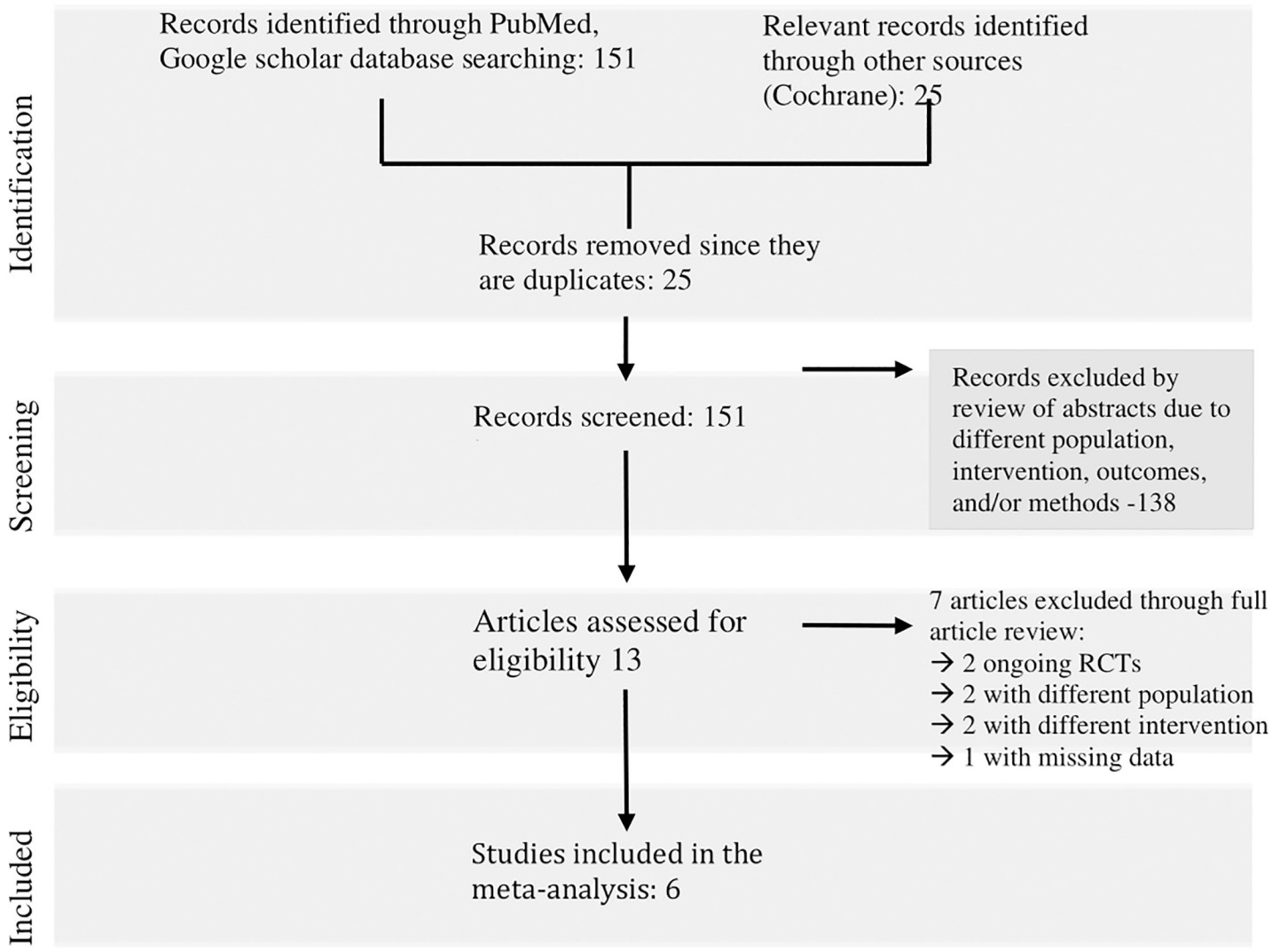

**Fig 1. Search strategy for identification of studies.**

study, correspondence with the author via email was done, but with no reply from the author until the time of writing. Six articles were subsequently included in the meta-analysis (Fig 1).

## Assessment of risk bias of included trials

Three independent reviewers extracted the data of interest using a standardized data collection form (S3 Appendix) and individually appraised each trial. The reviewers discussed the quality of included trials, outcomes to be collected, and risks of bias. Disparity in assessment was settled by an independent adjudicator. The assessment of random sequence generation, allocation concealment, incomplete outcome data, blinding of participants and personnel, blinding of outcome assessment, and intention-to-treat analysis was done using the quality scale for meta-analytic review, the Cochrane Collaboration Tool for Risk of Bias.

## Data analysis

Review Manager 5.3 was used to analyze the data. Analysis of dichotomous data was done using risk ratio, 95% confidence interval, and Mantel-Haenszel method with fixed effects

model when there was no evidence of statistical heterogeneity. A random effects model was used in the presence of heterogeneity. Heterogeneity between trials was tested using a standard Chi-square test and $I^2$ statistics. The p-value of <0.10 was considered to be statistically significant and $I^2$ of ≥50% is considered to have high heterogeneity.

## Description of studies

Six randomized controlled trials involving a total of 3,768 patients met the inclusion criteria. The invasive strategy group was composed of 1,986 patients while the conservative treatment group was consisted of 1,782 patients.

The data on population characteristics, intervention type, and measured outcomes were extracted from each trial (Table 1). The summary of numerical data extracted from each trial was illustrated in S4 Appendix. Four of the trials included patients with NSTEACS aged ≥ 70 years while two trials included patients ≥ 65 years old [10,15]. The studies compared the effectiveness of early invasive strategy (treatment group) versus optimum medical treatment (control group) in the management of NSTEACS in patients ≥ 65 years old.

In the control group all the trials used standard medical treatment [12–17]. In the treatment arm, four trials specified the time to intervention (4–72 hours) [12,14,15,16]. One of the six studies defined the time to intervention as coronary angiography and, if appropriate, revascularization within 7 days from admission for the index event [17]. Only one study did not specify the time to intervention but only mentioned "during initial admission" [13]. All of the trials included CABG as part of the intervention when indicated [12–17]. The study by Tegn et al. (2016) showed that 107 out of 229 (47%) underwent PCI while 6 out of 229 (3%) underwent CABG. Sanchis et al. (2016) reported that 28 out of 52 (54%) underwent PCI while 2 out of 52 (4%) underwent CABG. Savonnito et al. (2012) reported in their study that 76 out of 154 (50%) underwent PCI while 9 out of 154 (6%) underwent CABG. Bach et al. showed that 394 out of 679 (58%) underwent PCI while 285 out of 679 (42%) underwent CABG. Wallentin et al. (2016) reported that 926 out of 1,222 (76%) underwent revascularization during hospital admission for the index event but did not report what proportion underwent PCI and CABG. Puymirat et al. (2012) also reported that 860 out of 1,316 (65%) underwent PCI but also just stated that 71% had either PCI or CABG. Emails were sent to the authors to possibly obtain and clarify these information.

In the conservative group, a proportion of patients underwent coronary revascularization in 4 out of the 6 RCTs as follows: Bach: 404/1106 (37%), Sanchis: 5/54 (9%), Savonnito: 36/159 (23%), and Wallentin: 173/1235 (14%). In the remaining two studies, by Puymirat and Tegn, none of the patients in the conservative group underwent subsequent revascularization.

All trials assessed the outcome of all-cause mortality. Cardiovascular death was assessed by two trials [15,17]. Four trials reported the outcome of myocardial infarction [12,14,15,16]. Three trials assessed the outcome of stroke [12,14,15]. The outcomes of revascularization were reported by three studies [14–16]. The events of recurrent angina were assessed only by one study [15].

The Cochrane collaboration tool was used to assess the risk of bias. The random sequence generation, allocation concealment, incomplete outcome data, blinding of participants and personnel, blinding of outcome assessment, and intention-to-treat analysis were evaluated for each trial. All included trials were assessed to have low risk for bias (Table 2). Five funnel plots (Figs 2–6) for the presence of publication bias were illustrated in the following pages. There was no funnel plot for the outcome of recurrent angina. A symmetrical plot may suggest either a low probability or absence of publication bias. However, for our meta-analysis, the power of the funnel plot to detect true publication bias may be low because the trials included were less than 10.

**Table 1. Characteristics of included trials.**

| Study ID | Population | | Intervention | Outcome | Methods |
|---|---|---|---|---|---|
| Sanchis et al., 2016 N = 106 | **Inclusion:** Patients ≥ 70 years old with significant comorbidities diagnosed with NSTEMI **Mean age ± standard deviation (SD) of patients under the invasive strategy: 81 ± 5 Mean age ± SD of patients under the conservative strategy: 83 ± 6** | **Exclusion:** 1) Dynamic ST-segment changes; 2) Prior known non revascularizable CAD; 3) Concomitant heart disease different than ischemic heart disease; and 4) Life expectancy ≤1 year. | **Treatment Group:** Routine cardiac catheterization within 72 h of admission **Control Group:** Only medical treatment, although cardiac catheterization was allowed in the case of poor in-hospital outcome | **Primary:** Composite of all-cause mortality, recurrent myocardial infarction and readmission for cardiac cause **Secondary:** All-cause mortality, Reinfarction or Post-discharge revascularization, and bleeding episodes | Open label multicenter randomized controlled trial (Follow-up of 3 to 36 months) |
| Tegn et. al, 2016 N = 457 | **Inclusion:** Patients ≥ 80 years old with NSTEMI or Unstable Angina **Mean age (and range) of patients under the invasive strategy:84.7 (80–93) Mean age (and range) of patients under the conservative strategy:84.9 (80–94)** | **Exclusion:** 1) Clinically unstable; 2) Cardiogenic shock; 3) Continuing bleeding problems; or 4) Short life expectancy. | **Treatment Group:** Early coronary angiography (within 24 hours) with immediate assessment for adhoc PCI, CABG, or optimum medical treatment **Control Group:** Optimum medical treatment alone | **Primary:** Composite of MI, need for urgent revascularization stroke and death **Secondary:** Death from any cause | Open label multicenter randomized controlled trial (Follow-up of 3 years) |
| Wallentin et al., 2016 N = 2457 total patient population initially recruited **(N = 2421 patients with known survival status)** n = 1,292 total patient population with advanced age initially recruited **(n = 1,272 patients with advanced age with known survival status)** | **Inclusion:** Patients with suspicion of non-ST elevation acute coronary syndrome had to be verified by signs of ischemia with significant ST depression or pathological T-wave inversion on electrocardiography at rest, or by elevation of biochemical markers of myocardial damage. **Median age (and range) of patients under the invasive strategy[a]: 66 (40.8–84.5) Median age (and range) of patients under the conservative strategy[a]: 65.3 (37.5–83.5)** | **Exclusion:** Patients were excluded if they had indication, or had been treated within the past 24 h, for thrombolysis, had undergone angioplasty within the past 6 months, had had previous open-heart surgery, were at an advanced age (eg, older than 75 years; cutoff limits for advanced age varied between hospitals and countries and were decided by each unit), or had other conditions that made randomisation to early revascularisation inappropriate. | **Treatment Group:** Coronary angiography and, if appropriate, revascularisation within 7 days from admission for the index event ("Percutaneous coronary intervention was recommended in patients with one or two significant lesions whereas coronary artery bypass graft surgery was to be preferred in patients with three-vessel or left main disease.") **Control Group:** Coronary angiography in patients with refractory or recurrent symptoms despite optimum medical treatment, or severe ischemia as identified by a pre-discharge symptom-limited exercise test. | **Primary:** Composite of all-cause death and myocardial infarction **Secondary:** All-cause death, cardiac death, and new revascularisation procedures, hospital admissions for ischemic heart disease and any cardiac disease | Open label multicenter randomized controlled trial (Follow-up of 15 years) |
| Puymirat et al., 2012 N = 1,645 (total population) n = 658 (patient subgroup with age ≥ 75) | **Inclusion criteria:** Men or women aged over 18 years (Includes subgroup > 75 years old), who were admitted within 48 h after symptom onset for an acute MI **Mean age ± SD of patients under the invasive strategy:67.1 ±12.3 Mean age ± SD of patients under the conservative strategy: 79.7 ± 10.5** | **Exclusion:** 1) Iatrogenic MI; 2) ACS diagnosis invalidated in favor of another diagnosis; and 3) Patients with unstable angina and no increase in cardiac biomarkers. | **Treatment Group:** Early coronary angiography (during the initial hospital admission) **Control Group:** Received only medical therapy | **Primary:** Mortality, Minor bleeding, and Major bleeding | Open label multicenter randomized controlled trial (Follow-up of 3 years) |

(*Continued*)

**Table 1.** (Continued)

| Study ID | Population | | Intervention | Outcome | Methods |
|---|---|---|---|---|---|
| Savonnito, et al, 2012<br>N = 313 | **Inclusion:**<br>Patients ≥75 years old, assessed to have NSTEACS with cardiac ischemic symptoms at rest within 48 h<br>**Mean age ± SD of patients under the invasive strategy:81.8 ±4.4**<br>**Mean age ± SD of patients under the conservative strategy: 81.8 ± 4.7** | **Exclusion:**<br>1) Secondary causes of myocardial ischemia;<br>2) Ongoing myocardial ischemia or heart failure despite optimized therapy;<br>3) PCI or CABG within 30 days before randomization;<br>4) Serum creatinine >2.5 mg/dl;<br>5) Cerebrovascular accident within the previous month;<br>6) Recent transfusions;<br>7) Gastrointestinal or genitourinary bleeding within 6 weeks before randomization;<br>8) Platelet count <90,000 cells/ul<br>9) Ongoing oral anticoagulation<br>10) Severe obstructive lung disease<br>11) Malignancy;<br>12) Neurological deficit limiting follow-up. | **Treatment group:**<br>Coronary angiography within 72 h and, when indicated, coronary revascularization by either PCI or CABG<br>**Control Group:**<br>Initially conservative therapy and coronary angiography during index hospital stay was allowed in the case of refractory ischemia, myocardial (re)infarction, heart failure of ischemic origin, or malignant ventricular arrhythmias | **Primary:**<br>Composite of all-cause mortality, non-fatal MI, disabling stroke, and repeat hospital stay for cardiovascular causes or severe bleeding within 12 months | Open randomized controlled trial (Follow-up of 1 year) |
| Bach et al., 2004<br>N = 2, 220 (total population)<br>n = 962 (patient subgroup with age ≥ 65) | **Inclusion:**<br>Patients older than 18 years of age (with subgroup of ≥ 65 years old) with episode of angina in the preceding 24 hours; Candidates for coronary revascularization<br>**Mean age ± SD of the subgroup ≥ 65 years old: 72.9 ± 5.6** | **Exclusion:**<br>1) Persistent ST-segment elevation; 2) Secondary angina;<br>3) Percutaneous coronary revascularization or coronary bypass surgery within the previous 6 months; 4) Unstable comorbidities;<br>5) Left bundle-branch block or paced rhythm;<br>6) Severe congestive heart failure or cardiogenic shock;<br>7) Clinically important systemic disease;<br>8) Serum creatinine concentration greater than 220 umol/L (>2.5 mg/dL);<br>9) Treatment with a glycoprotein IIb/IIIa antagonist within the past 96 hours; or 10) Ongoing long-term treatment with ticlopidine, clopidogrel, or warfarin. | **Treatment Group:** Coronary angiography 4 to 48 hours after randomization (cardiac catheterization, PCI, and CABG surgery during the index hospitalization)<br>**Control Group:**<br>Medical treatment; Coronary angiography was reserved for patients who had certain high-risk characteristics consistent with failure of medical therapy or stress-induced ischemia | **Primary:**<br>Rates of 30-day and 6-month mortality, nonfatal MI, rehospitalization, stroke, and hemorrhagic complications | Open randomized controlled trial (Follow-up of 6 months and 1 year) |

[a]Median age of the subgroup with age > 65 years old was not reported in the study of Wallentin et al.

# Results

## Effects of intervention on outcomes of interest

**A. All-cause mortality.** The follow-up periods of each RCT in this study were summarized in Table 1. For the outcome of all-cause mortality, the mean duration of follow-up is 4

**Table 2. Quality assessment table.**

| Study ID | Method of Random Sequence Generation (Selection Bias) | Method of Allocation Concealment (Selection Bias) | Incomplete Outcome Data/Loss of participants to follow up (Attrition Bias) | Blinding of Participants and Personnel (Performance Bias) | Blinding of Outcome Assessment (Detection Bias) | Selective Reporting/ Intention to treat analysis (Reporting Bias) |
|---|---|---|---|---|---|---|
| Sanchis et al., 2016 | Low Risk | Low Risk | Low Risk | Low Risk | Low Risk | Low Risk |
| Tegn et. Al, 2016 | Low Risk | Low Risk | Low Risk | Low Risk | Low Risk | Low Risk |
| Wallentin et al., 2016 | Low Risk | Low Risk | Low Risk | Low Risk | Low Risk | Low Risk |
| Puymirat et al., 2012 | Low Risk | Low Risk | Low Risk | Low Risk | Low Risk | Low Risk |
| Savonnito, et al, 2012 | Low Risk | Low Risk | Low Risk | Low Risk | Low Risk | Low Risk |
| Bach et al., 2004 | Low Risk | Low Risk | Low Risk | Low Risk | Low Risk | Low Risk |

years. A total of 607 among 1,986 (31%) patients ≥ 65 years old with NSTEACS died in the Invasive Strategy Group; while 646 died among 1,782 (36%) patients in the Conservative Group (Fig 7). The pooled analysis of all-cause mortality showed no significant effect of

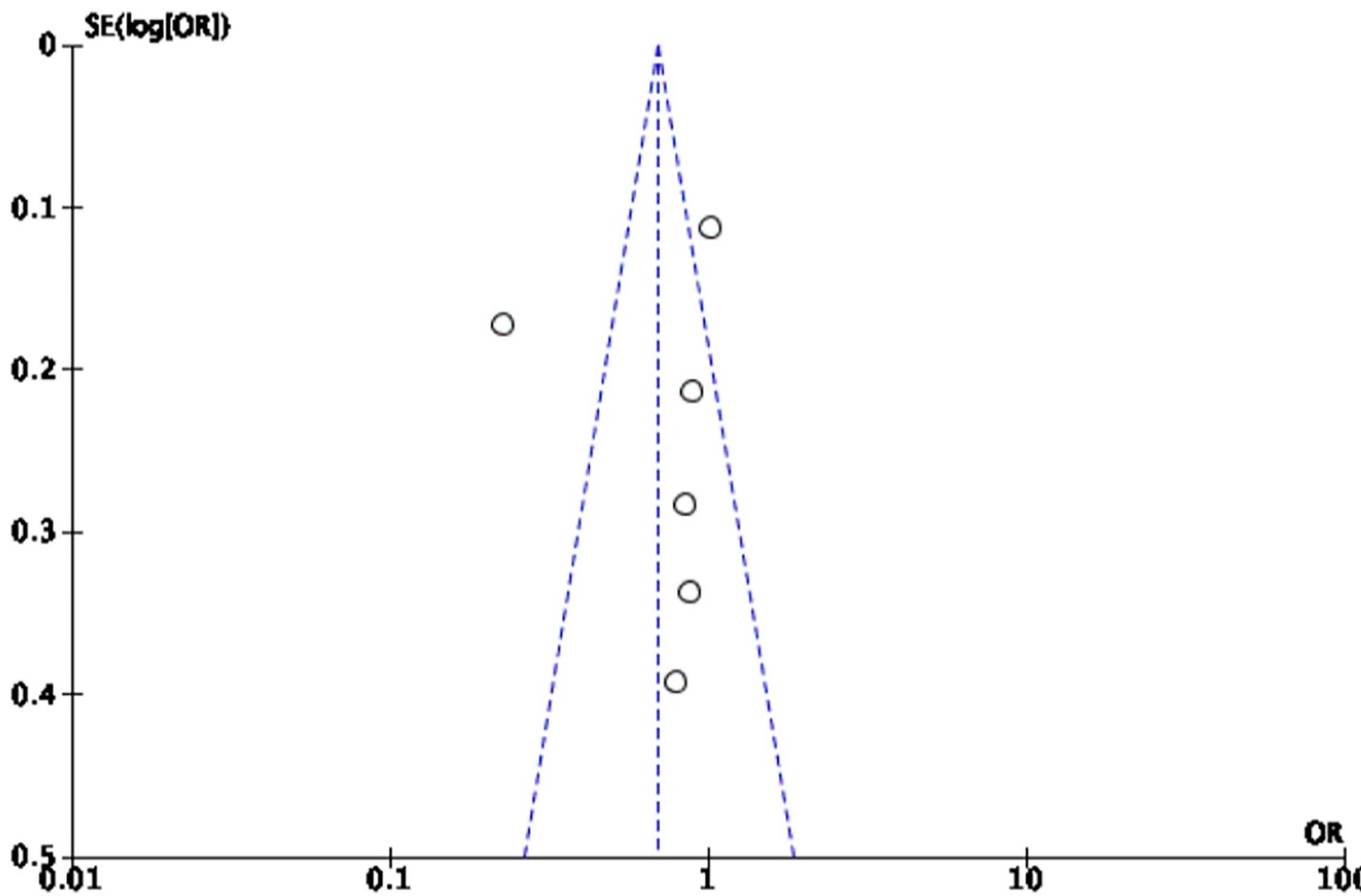

**Fig 2. Funnel plot for the assessment of publication bias for the six randomized controlled trials that assessed the outcome of all-cause mortality.**

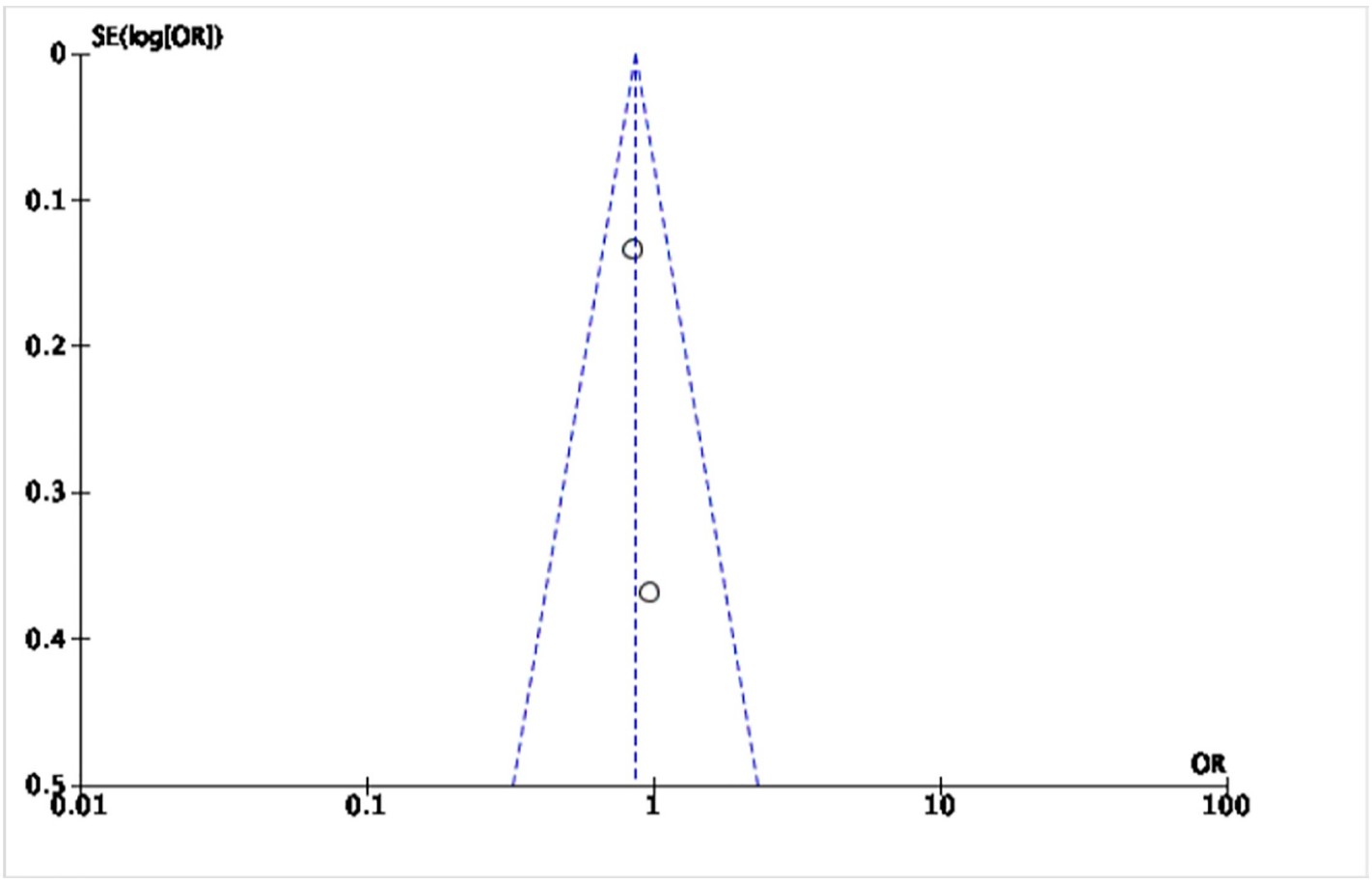

**Fig 3. Funnel plot for the assessment of publication bias for the two randomized controlled trials that assessed the outcome of cardiovascular mortality.**

invasive strategy on the outcome with an overall risk ratio of 0.69 (95% CI 0.39 to 1.23) with significant heterogeneity (p value of 0.00001, $I^2$ = 91%) using the Random Effects model. Initial analysis using the Fixed Effects model was illustrated in Fig 8. Analysis showed that the study by Puymirat et al. achieved results that were strongly different from other studies in terms of the outcome of all-cause mortality. Hence, a pooled analysis that excluded this study was attempted but the results were still not significant as illustrated in Fig 9.

**B. Cardiovascular mortality.** The mean duration of follow-up for cardiovascular mortality was 8 years. A total of 156 among 802 (19%) patients ≥ 65 years old with NSTEACS developed the outcome of cardiovascular mortality in the Invasive Strategy Group; while 170 among 783 (22%) patients died of cardiovascular cause in the Conservative Group (Fig 10). The pooled analysis of cardiovascular mortality showed no significant effect of invasive strategy on the outcome with an overall risk ratio of 0.86 (95% CI 0.67 to 1.10).

**C. Myocardial infarction.** The average follow-up for the outcome of MI was 2 years. In the Invasive Strategy Group, there were 89 events of MI among a total of 926 (10%) patients; while there were 142 among 912 (16%) patients in the Conservative Group (Fig 11). The pooled analysis showed that invasive strategy showed no significant effect on the outcome versus conservative treatment in preventing MI with an overall risk ratio of 0.63 (95% CI 0.39 to 1.04) with significant heterogeneity (p value of 0.07, $I^2$ = 60%) using the

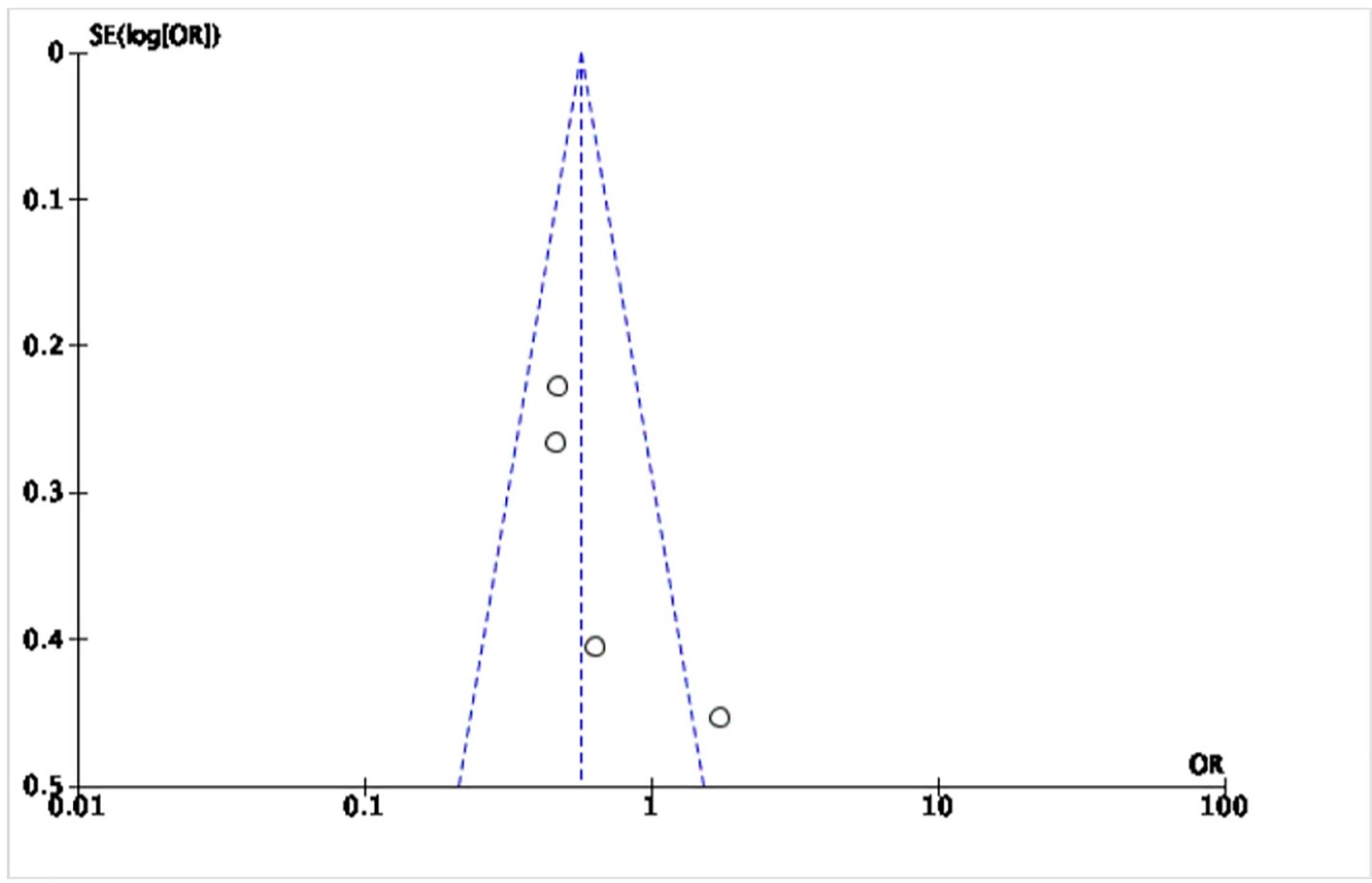

**Fig 4. Funnel plot for the assessment of publication bias for the four randomized controlled trials that assessed the outcome of myocardial infarction.**

Random Effects model. An initial analysis using the Fixed Effects model was illustrated in Fig 12.

**D. Stroke.**   Similar to MI, the mean follow-up for the outcome of stroke was 2 years. Among the six trials, Savonitto et al., Tegn et al., and Bach et al. reported the outcomes of stroke (Fig 13). In the Invasive Strategy Group, there were 13 events of stroke among 874 (2%) patients; while there were 24 among 858 (3%) patients in the Conservative Group. The pooled analysis showed that early invasive strategy showed no significant effect on outcomes of stroke with overall risk ratio of 0.52 (95% CI 0.26–1.03, $I^2 = 0\%$).

**E. Need for revascularization.**   The average duration of follow-up for this outcome was also 2 years. In elderly patients with NSTEACS, there were a total of 10 patients among 435 (2%) who needed revascularization in the Invasive Group while there were 34 patients among 441 (8%) in the Conservative Group (Fig 14). The pooled analysis for need for revascularization showed statistically significant benefit with an overall risk ratio of 0.29 (95% CI 0.14 to 0.59) with no significant heterogeneity (p value of 0.0006, $I^2 = 3\%$).

**F. Outcome for recurrent angina.**   Among the six trials, only one trial assessed the outcome of recurrent angina [15]. The follow-up period for this outcome was 1 year. An invasive strategy showed no significant effect on the outcomes of recurrent angina (RR 0.81, 95% CI 0.45–1.46, p = 0.49).

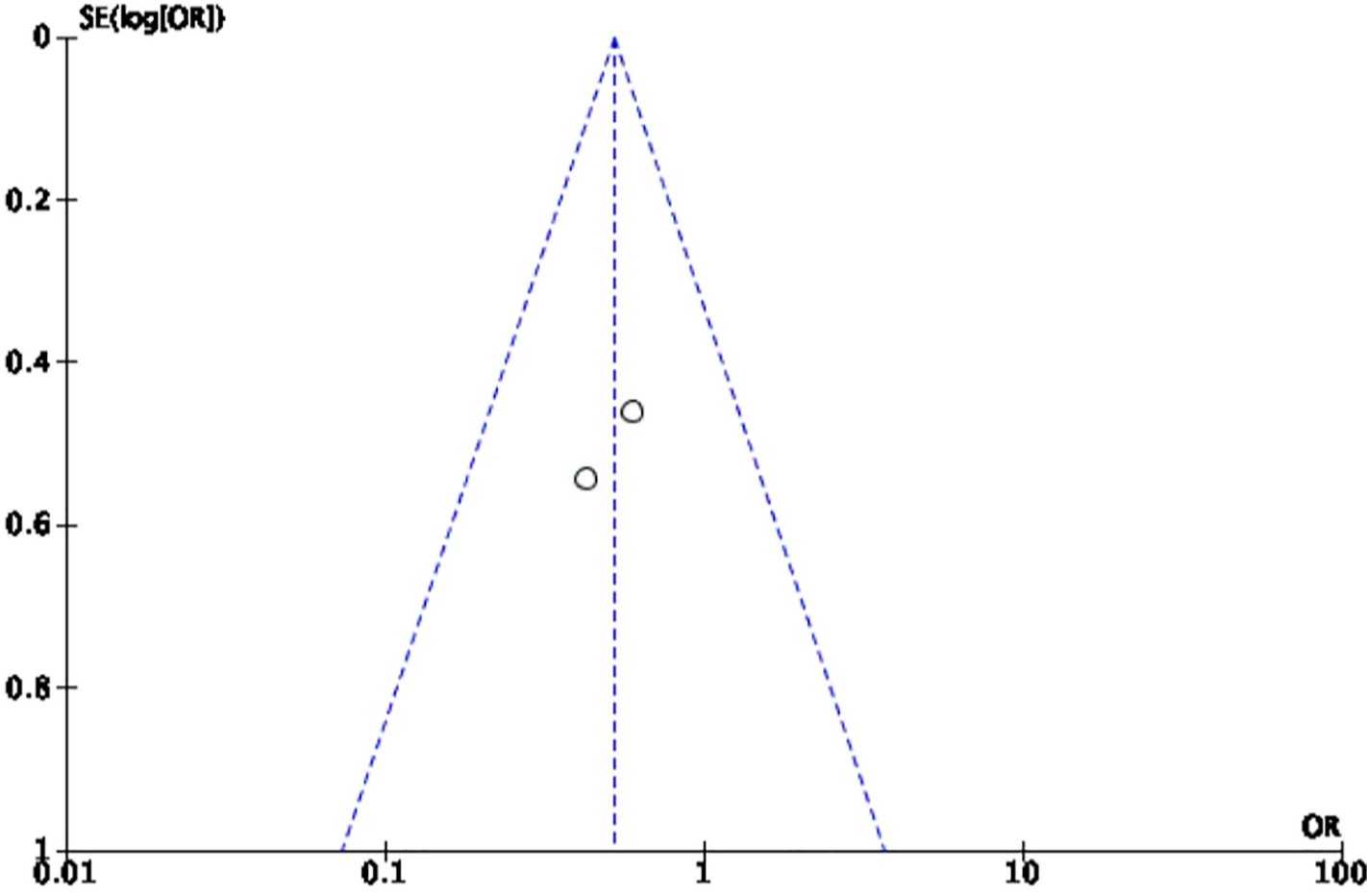

**Fig 5. Funnel plot for the assessment of publication bias for the three randomized controlled trials that assessed the outcome of stroke.** (One study had zero outcomes of stroke in both interventions).

## Discussion

Meta-analysis of data from the six trials included in this study showed that an early invasive strategy appears to be beneficial in terms of reduction of need for revascularization in suitable patients $\geq$ 65 years old with NSTEACS. This finding implies that more patients in the conservative group clinically worsened during their course in the ward, requiring revascularization. It is also possible that early anatomic definition of the diseased coronaries may help the attending physician optimize an appropriate evidence-based management of the patient. The studies that evaluated the outcomes of revascularization stated that the indications for revascularization in the conservative group were: positive pre-discharge stress test, poor in-hospital outcomes, recurrent ischemia, reinfarction, malignant ventricular arrhythmias, refractory angina, and heart failure [14–16]. Some patients who subsequently required revascularization could have probably been better off with an early invasive approach. However, it is important to note that although there was a significant difference in the need for revascularization between the invasive (2%) and conservative (8%) groups, the rates of revascularization for both groups are low; hence, an implication that an invasive strategy is not a commonly applied management for this population limiting its broad application.

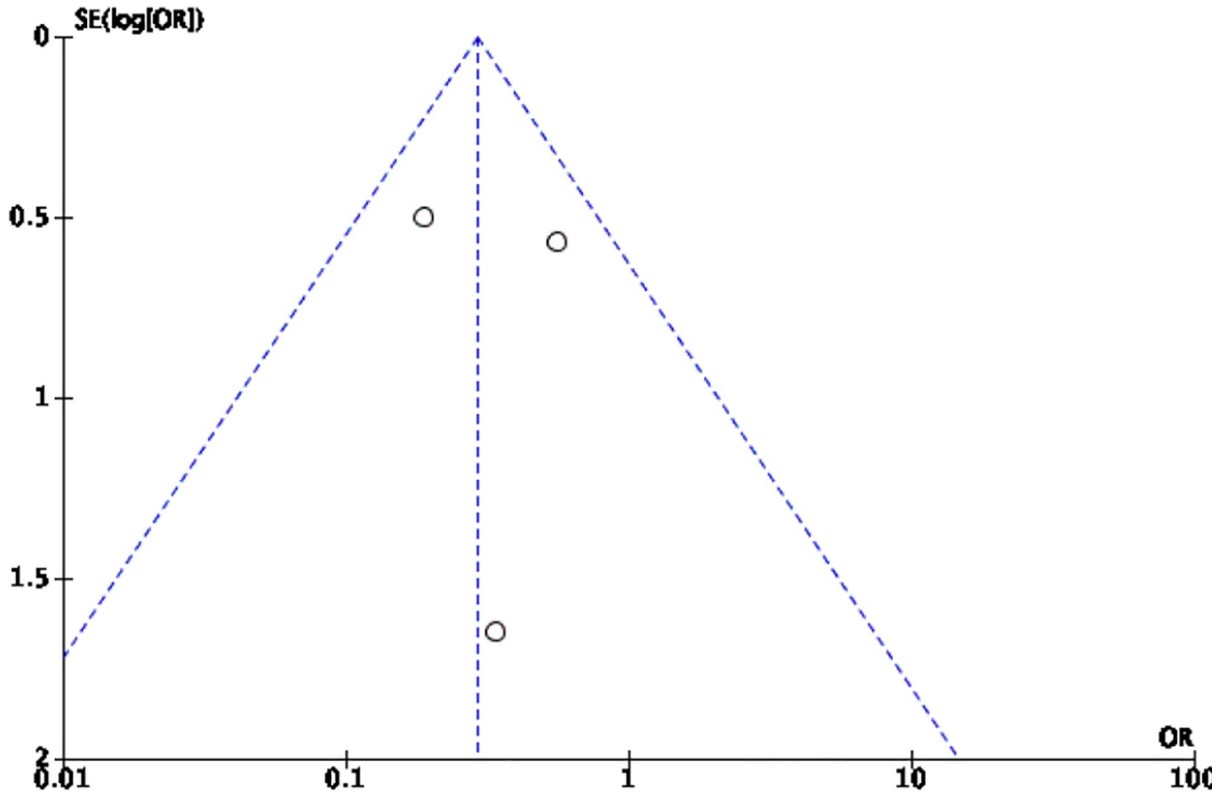

**Fig 6. Funnel plot for the assessment of publication bias for the three randomized controlled trials that assessed the outcome of need for revascularization.**

For the outcomes of death and MI, an invasive strategy showed no significant effect on the outcomes versus conservative treatment with significant heterogeneity. The possible sources of heterogeneity for the outcomes of death and MI may be the small number of events and sample sizes. In three out of the six studies, the patient population $\geq$ 65 years old was just a subgroup analysis of the total population [12,13,17]. Hence, the population in the subgroup analysis may not be powered enough to detect the differences in the intervention and

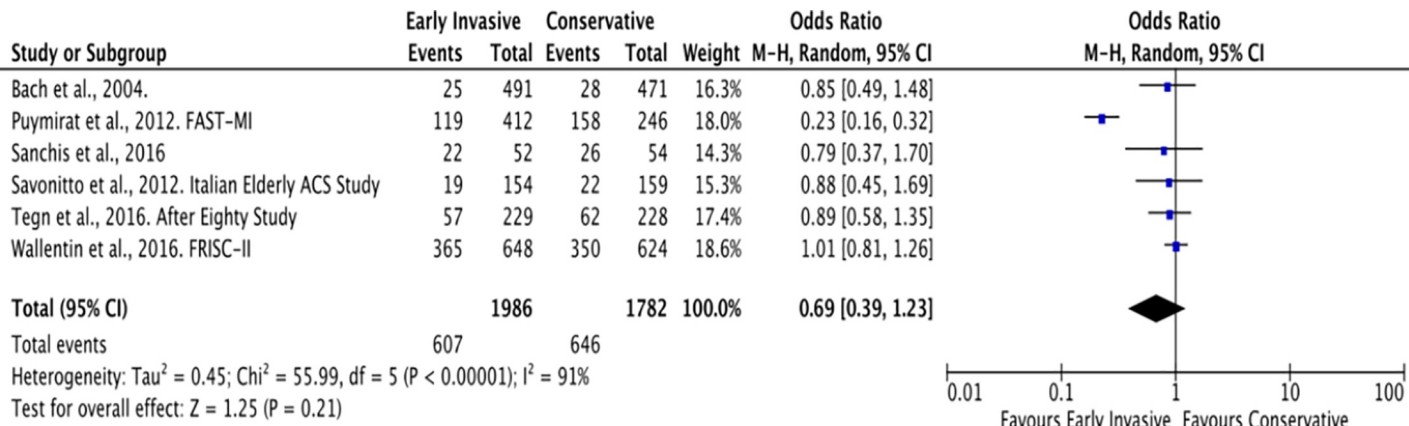

**Fig 7. Comparison between invasive and conservative strategy with the outcome of all-cause mortality.**

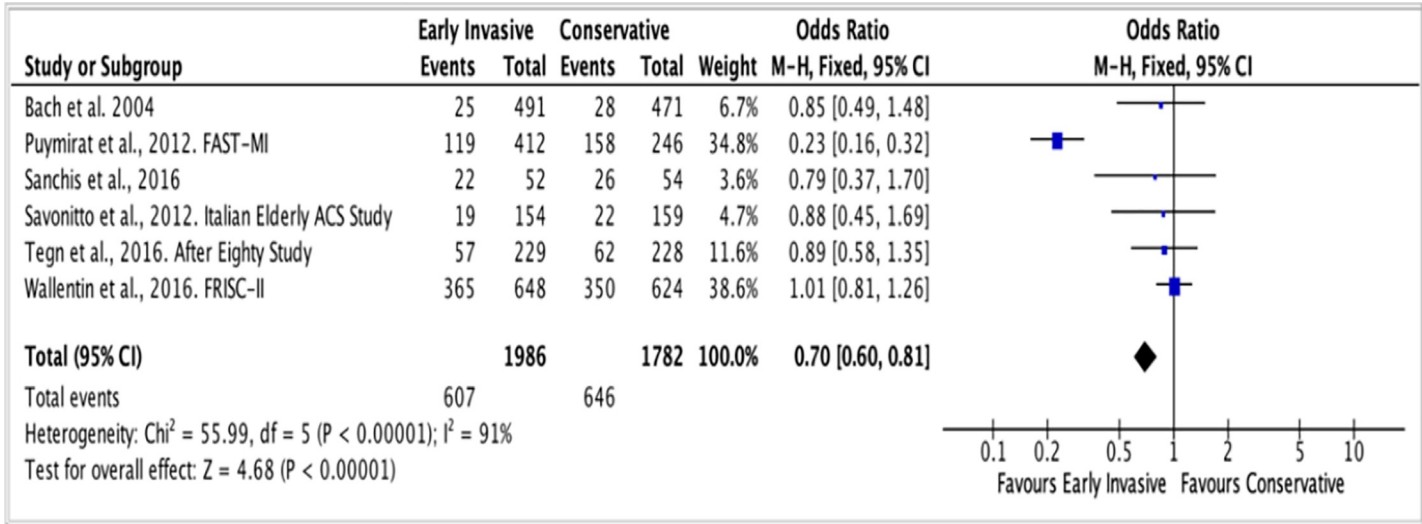

**Fig 8. Comparison between invasive and conservative strategy with the outcome of all-cause mortality using Fixed Effects model.**

outcomes of interest. In addition, there were differences in age cutoffs. Two studies had age cutoffs of 75 years [12,15]; two had cutoffs of 65 years [12,17]; while the remaining two had age cutoffs of 70 and 80 years [14,16]. Possible clinical differences in outcomes may exist within the age brackets of this particular population. Furthermore, there were differences in follow-up periods, ranging from 3 months to 15 years, which may have contributed to the heterogeneity since shorter follow-up would mean lesser chance of catching the outcome of interest but longer follow-up would mean higher probability of the event occurring [12–17]. Likewise, the application of invasive strategy did not lead to reduction of cardiovascular mortality and stroke. Recurrent angina was assessed only in one study [15], which also showed no significant findings in favor of invasive strategy.

Overall, this study does not support the relatively conservative tendency when dealing with patients ≥ 65 years old with NSTEACS in real-life clinical setting. The advance-aged population is considered a high-risk group wherein more than half the mortality in NSTEACS occur

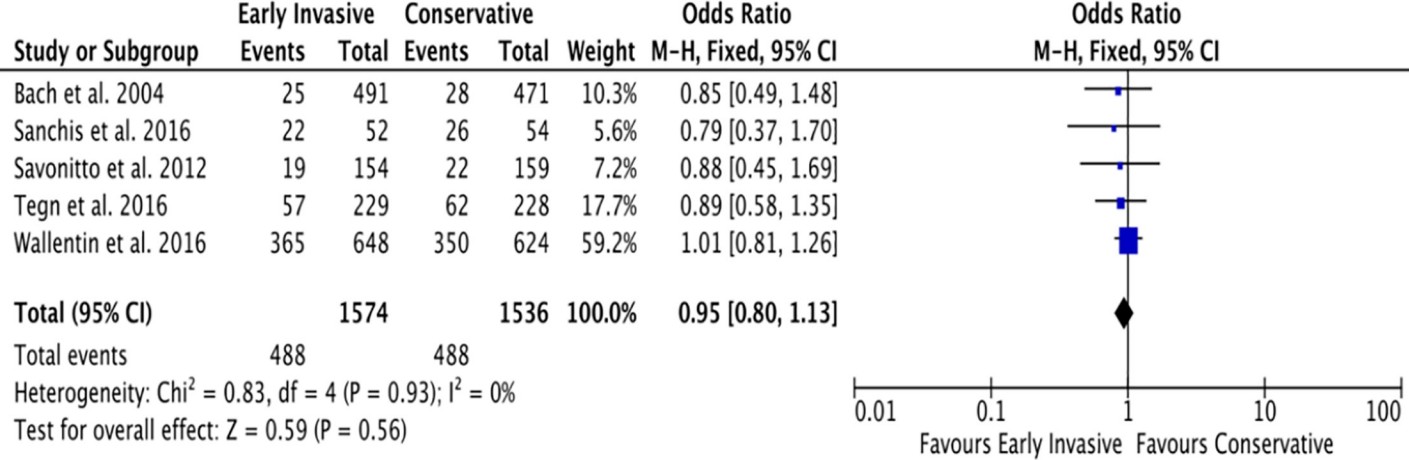

**Fig 9. Pooled analysis of 5 out of 6 RCTs (excluding the study by Puymirat et al.) showing comparison between invasive and conservative strategy with the outcome of all-cause mortality using the Fixed Effects model.**

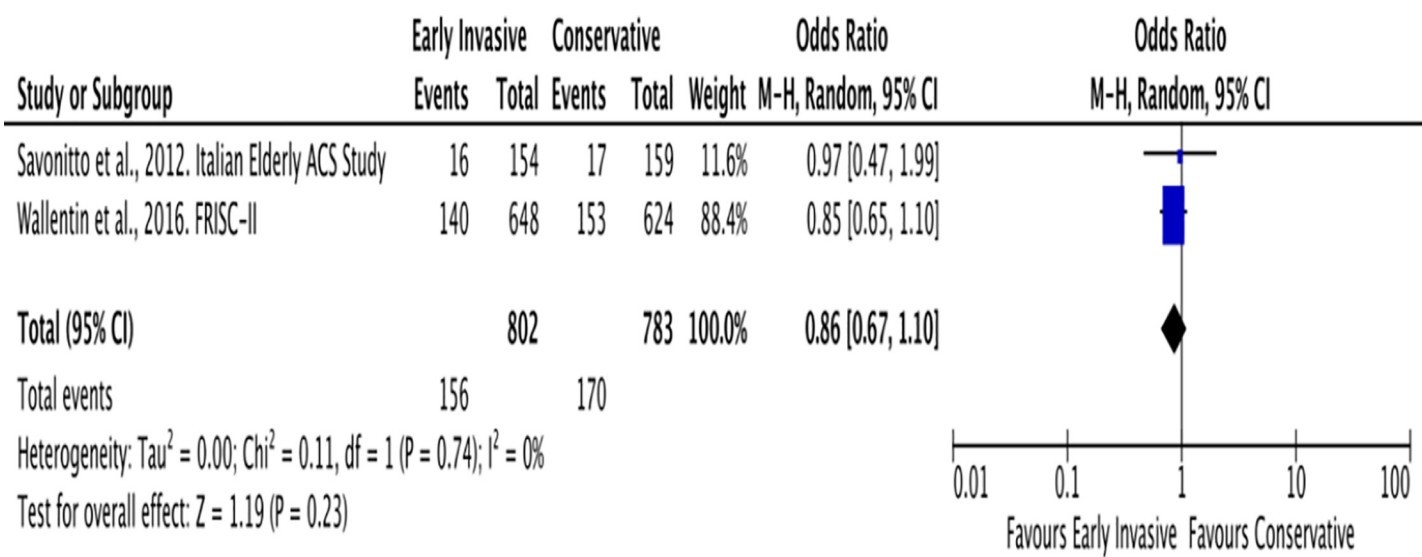

**Fig 10. Comparison between invasive and conservative strategy with the outcome of cardiovascular mortality.**

and a more aggressive approach in suitable patients may be more appropriate and beneficial particularly in reducing need for revascularization [5].

Frailty is common among older adults and is defined as a "state of reduced physiological reserve and increased vulnerability for poor resolution of homeostasis after a stressor event" [18]. A recent meta-analysis showed that frailty was associated with increased risk of cardio-vascular diseases and was also associated with a 3-fold increase in risk for cardiovascular death [19]. The mechanism was said to be multi-factorial, likely owing to vascular, cardiac, cellular, bio-humoral, and endocrine alterations in the frail condition [19]. In addition, most especially during an ACS that is a stressor event, the increase in inflammation and thrombosis places the older patients at a more vulnerable state. Hence, a more aggressive approach may be suitable for appropriate, high cardiovascular risk patients.

| Study or Subgroup | Early Invasive Events | Total | Conservative Events | Total | Weight | Odds Ratio M-H, Random, 95% CI |
|---|---|---|---|---|---|---|
| Bach et al., 2004. | 23 | 491 | 45 | 471 | 29.3% | 0.47 [0.28, 0.78] |
| Sanchis et al., 2016 | 16 | 52 | 11 | 54 | 18.1% | 1.74 [0.72, 4.21] |
| Savonitto et al., 2012. Italian Elderly ACS Study | 11 | 154 | 17 | 159 | 20.5% | 0.64 [0.29, 1.42] |
| Tegn et al., 2016. After Eighty Study | 39 | 229 | 69 | 228 | 32.1% | 0.47 [0.30, 0.74] |
| Total (95% CI) | | 926 | | 912 | 100.0% | 0.63 [0.39, 1.04] |
| Total events | 89 | | 142 | | | |

Heterogeneity: Tau$^2$ = 0.15; Chi$^2$ = 7.42, df = 3 (P = 0.06); I$^2$ = 60%
Test for overall effect: Z = 1.81 (P = 0.07)

**Fig 11. Comparison between invasive and conservative strategy with the outcome of myocardial infarction.**

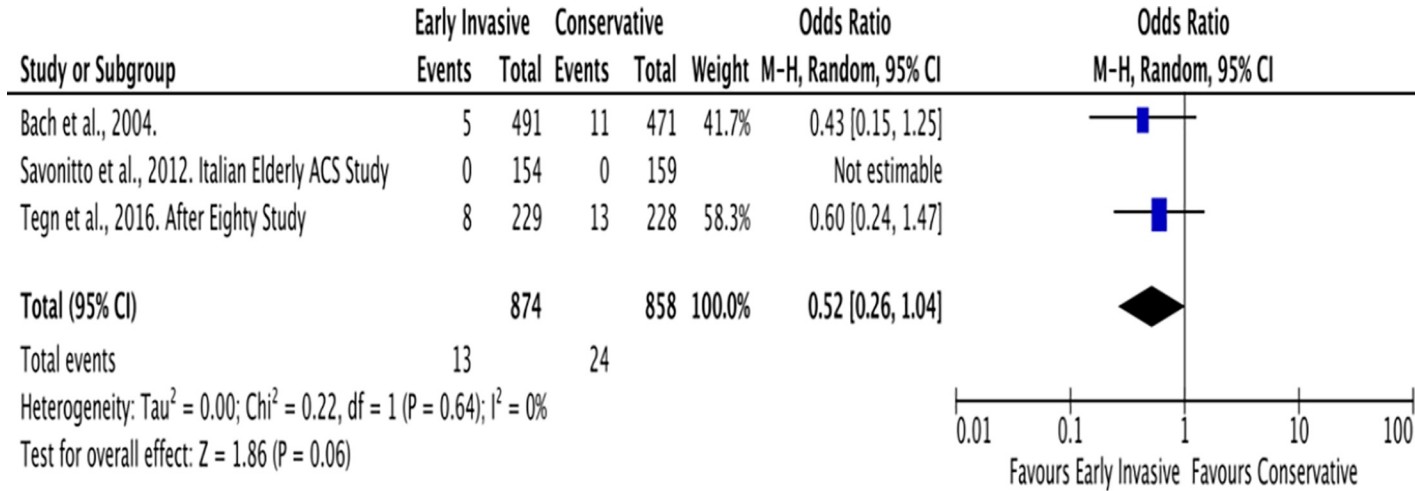

| Study or Subgroup | Early Invasive Events | Total | Conservative Events | Total | Weight | Odds Ratio M-H, Fixed, 95% CI | Odds Ratio M-H, Fixed, 95% CI |
|---|---|---|---|---|---|---|---|
| Bach et al., 2004. | 23 | 491 | 45 | 471 | 35.3% | 0.47 [0.28, 0.78] | |
| Sanchis et al., 2016 | 16 | 52 | 11 | 54 | 6.0% | 1.74 [0.72, 4.21] | |
| Savonitto et al., 2012. Italian Elderly ACS Study | 11 | 154 | 17 | 159 | 12.5% | 0.64 [0.29, 1.42] | |
| Tegn et al., 2016. After Eighty Study | 39 | 229 | 69 | 228 | 46.2% | 0.47 [0.30, 0.74] | |
| Total (95% CI) | | 926 | | 912 | 100.0% | 0.57 [0.42, 0.76] | |
| Total events | 89 | | 142 | | | | |

Heterogeneity: Chi$^2$ = 7.42, df = 3 (P = 0.06); I$^2$ = 60%
Test for overall effect: Z = 3.83 (P = 0.0001)

**Fig 12. Comparison between invasive and conservative strategy with the outcome of myocardial infarction using the Fixed Effects model.**

Among people who die of ischemic heart disease, 83% were >65 years of age [1]. This mortality rate is expected to increase in the forthcoming decades due to improving life expectancy of the elderly. Age is one of the most important predictors of risk in NSTEACS. Each 10-year increase in age results in a 75% increase in hospital mortality in ACS patients [20]. Despite the relatively higher risk in this age group, older ACS patients are under-represented in clinical trials such that subjects older than 75 years of age account for less than 10%, and those older than 85 years account for less than 2% of all NSTEACS subjects [7]. This highlights the need for more clinical trials and studies in this age group.

Data from the CRUSADE (Can Rapid Risk Stratification of Unstable Angina Patients Suppress Adverse Outcomes with Early Implementation of the American College of Cardiology/American Heart Association Guidelines) registry showed that NSTEMI patients aged ≥ 65 years who experienced an in-hospital major bleed had a 33% increased risk of 30-day mortality [21]. However, the advancement of equipment and technique has made PCI safer for even

| Study or Subgroup | Early Invasive Events | Total | Conservative Events | Total | Weight | Odds Ratio M-H, Random, 95% CI | Odds Ratio M-H, Random, 95% CI |
|---|---|---|---|---|---|---|---|
| Bach et al., 2004. | 5 | 491 | 11 | 471 | 41.7% | 0.43 [0.15, 1.25] | |
| Savonitto et al., 2012. Italian Elderly ACS Study | 0 | 154 | 0 | 159 | | Not estimable | |
| Tegn et al., 2016. After Eighty Study | 8 | 229 | 13 | 228 | 58.3% | 0.60 [0.24, 1.47] | |
| Total (95% CI) | | 874 | | 858 | 100.0% | 0.52 [0.26, 1.04] | |
| Total events | 13 | | 24 | | | | |

Heterogeneity: Tau$^2$ = 0.00; Chi$^2$ = 0.22, df = 1 (P = 0.64); I$^2$ = 0%
Test for overall effect: Z = 1.86 (P = 0.06)

**Fig 13. Comparison between invasive and conservative strategy with the outcome of stroke.**

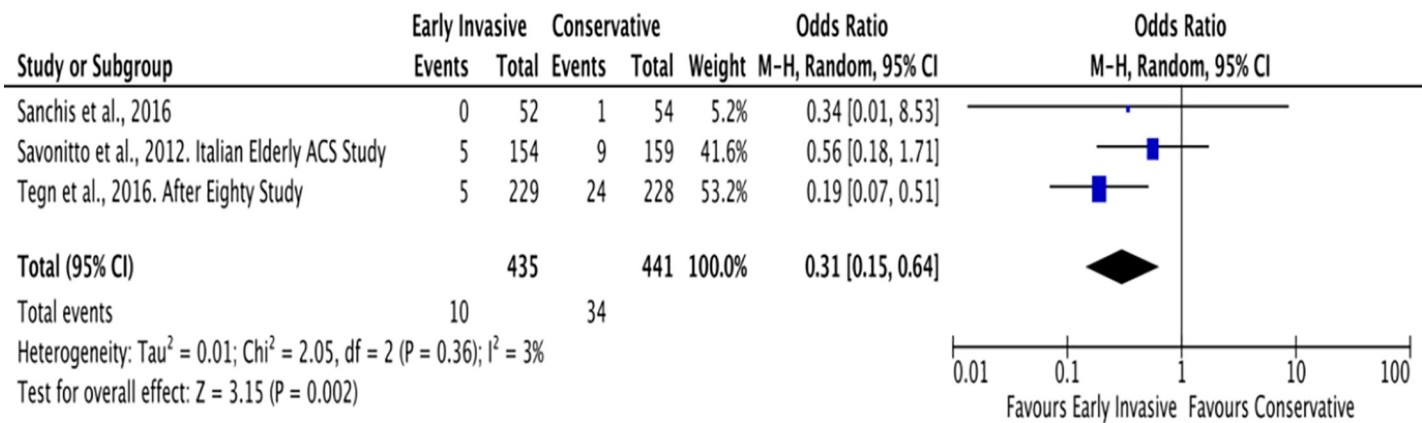

**Fig 14. Comparison between invasive and conservative strategy with the outcome of need for revascularization.**

very elderly patients ($\geq$ 90 years of age) with high success rates and declining major bleeding risk [22].

The length of dual antiplatelet therapy (DAPT) for older patients is a concern especially for those who underwent PCI. However, there are still limited data on the optimal DAPT duration for patients for whom bleeding and ischemic risks should be considered. A recent cohort study (data obtained from the RENAMI Registry) evaluated the benefit-to-risk ratio of different DAPT durations (< 12 months, 12 months, and > 12 months) in ACS patients who underwent PCI [21]. The study showed that DAPT of at least 12 months is more beneficial than shorter DAPT in reducing net adverse clinical events (NACE), a combination of MACE and major bleeding complications. However, the result was different for the subgroup population > 75 years of age in this study since a reduced benefit was noted for prolonged DAPT > 12 months. The higher risk for harm than benefit was noted for this particular group most likely due to impaired renal function, peri-, and post-procedural bleeding [21]. Nevertheless, this issue may be addressed by using "PRECISE-DAPT" Score, a simple five-item tool that guides the optimal DAPT duration after PCI [22]. The study recommended that their finding of less favorable risk-benefit ratio for prolonged DAPT in those > 75 years old should be confirmed by RCTs.

## Summary and conclusion

Results of this meta-analysis showed that there was a significant reduction in the need for revascularization in the invasive strategy group compared to the conservative treatment group in NSTEACS patients $\geq$ 65 years old. This finding does not support the bias against early routine invasive intervention in this patient population.

Although an early invasive strategy may be favorable in the need for revascularization among those $\geq$ 65 years old presenting with NSTEACS, the certainty of benefit versus risk still needs to be supported by larger clinical trials and registries with uniform age cutoff for this population, particularly $\geq$ 65 years old, to provide high generalizability and statistical power. Current risk scoring systems such as the GRACE (Global Registry of Acute Coronary Events) Score, TIMI (Thrombolysis in Myocardial Infarction) Risk Score, and CRUSADE Bleeding Score are recommended in the initial evaluation of elderly patients presenting with NSTEACS. A special risk scoring may be developed to more accurately identify those who are suitable for an early invasive strategy, with an expected larger outcome and survival benefit.

## Supporting information

**S1 Checklist.**
(DOCX)

**S1 Appendix. PubMed search strategy.**
(XLSX)

**S2 Appendix. Excluded studies and reasons for exclusion.**
(XLSX)

**S3 Appendix. Sample data extraction template.**
(DOCX)

**S4 Appendix. Summary of results of the six included randomized controlled trials.**
(DOCX)

## Acknowledgments

The authors would like to thank God, their family, and friends for their support and encouragement in making this meta-analysis possible. The authors are also extending their gratitude to the administration of Manila Doctors Hospital (MDH); the chairman of the Department of Internal Medicine, Dr. Petrarch B. Bravo; the head of the research committee of the Department of Internal Medicine, Dr. Elmer Jasper L. Llanes; and the chairman and training officer of the MDH Section of Cardiology, Dr. Nelson S. Abelardo and Dr. Rogelio V. Tangco, for always being supportive in the endeavors of their trainees.

## Author Contributions

**Conceptualization:** Joan Dymphna P. Reaño, Karen V. Miralles, Maria Grethel C. Dimalala, Rafael R. Castillo.

**Data curation:** Joan Dymphna P. Reaño, Louie Alfred B. Shiu, Karen V. Miralles, Maria Grethel C. Dimalala.

**Formal analysis:** Joan Dymphna P. Reaño, Louie Alfred B. Shiu, Karen V. Miralles, Maria Grethel C. Dimalala, Noemi S. Pestaño, Felix Eduardo R. Punzalan, Bernadette Tumanan-Mendoza, Michael Joseph T. Reyes.

**Investigation:** Joan Dymphna P. Reaño, Louie Alfred B. Shiu, Rafael R. Castillo.

**Methodology:** Joan Dymphna P. Reaño, Louie Alfred B. Shiu, Karen V. Miralles, Maria Grethel C. Dimalala.

**Project administration:** Joan Dymphna P. Reaño.

**Software:** Maria Grethel C. Dimalala.

**Supervision:** Joan Dymphna P. Reaño, Louie Alfred B. Shiu, Karen V. Miralles, Noemi S. Pestaño, Felix Eduardo R. Punzalan, Bernadette Tumanan-Mendoza, Michael Joseph T. Reyes.

**Validation:** Joan Dymphna P. Reaño, Karen V. Miralles, Noemi S. Pestaño, Felix Eduardo R. Punzalan, Bernadette Tumanan-Mendoza, Michael Joseph T. Reyes.

**Visualization:** Joan Dymphna P. Reaño.

**Writing – original draft:** Joan Dymphna P. Reaño, Rafael R. Castillo.

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
