## [Decision Letter · Decision Letter 0]

19 Nov 2019

PONE-D-19-21413

A systematic review and meta-analysis on the effectiveness of an early invasive strategy compared to a conservative approach in elderly patients with non-ST elevation acute coronary syndrome

PLOS ONE

Dear Dr. Reano,

Thank you for submitting your manuscript to PLOS ONE. After careful consideration, we feel that it has merit but does not fully meet PLOS ONE’s publication criteria as it currently stands. Therefore, we invite you to submit a revised version of the manuscript that addresses the points raised during the review process.

We would appreciate receiving your revised manuscript by Jan 03 2020 11:59PM. To enhance the reproducibility of your results, we recommend that if applicable you deposit your laboratory protocols in protocols.io, where a protocol can be assigned its own identifier (DOI) such that it can be cited independently in the future. For instructions see: http://journals.plos.org/plosone/s/submission-guidelines#loc-laboratory-protocols

We look forward to receiving your revised manuscript.

Kind regards,

Carmine Pizzi

Academic Editor

PLOS ONE

1. Thank you for inlcuding your competing interests statement; "I have read the journal's policy and the authors of this manuscript have the following competing interests:

RRC: member of advisory board or speakers’ pool of Servier, Boehringer Ingelheim, Menarini, LRI-Therapharma, Sanofi, UAP Pharma, Unilab; MTR: member of speakers’ pool of Novartis, Servier, Astra Zeneca; the rest declare no conflict of interest."

2. Please ensure that you refer to Figures 7-11 in your text as, if accepted, production will need this reference to link the reader to the figure.

Review Comments to the Author

Reviewer #1: The paper is very interesting.

Major issues.

Elderly patients in the title and abstract and the whole paper should be better defined>older than 65 years old is not "elderly". Maybe adding median age should be better

Abstract>in the methods authors state that they used fixed effect, although with high inconstency (i2 90%). Fixed effect should not be used for levels of I2 more than 30%. Please follow cochrane guidelines.

Frailty should be largely discusse, as may deeply impact on management of these patients (quote PMID: 28143778 ).

Lenght of dapt in these patients remains an issue, and should

1) explored with meta. regression analysis

2) discussed (quote on PMID: 30862233)

Reviewer #2: This metanalysis would basically be helpful as it tries to clarify a relevant point about management of NSTE-ACS elderly patients.

The following points limit, however, the conclusions deriving from the analysis.

1. Some more information would be helpful to put the results in the right perspective. Thus, the authors should: a) report the average duration of follow-up for each of the end-points; b) to specify the proportion of patients undergoing by-pass surgery vs. percutaneous coronary intervention; c) the proportion of patients who underwent coronary revascularization during hospital staying in the conservative group.

2. Figure 2 shows that the study by Puymirat et al. achieved results strongly different from the other studies as far as global mortality is concerned. Accordingly, the authors should at least present the results of the metanalysis also after removing this study.

3. Although significantly higher than that of the interventional group, coronary revascularization in the conservative group was rather low (8% vs. 2%), which may not justify a broad application of the invasive strategy to all elderly patients.

---

## [Author Response · Author response to Decision Letter 0]

12 Jan 2020

Dear Ms. Mentsl,

Attached here is my PRISMA CHECK LIST as requested in your latest email to me.

With regards to the competing interests, in page 27 of the manuscript, under the "Declaration of conflicts of interests", we have already added this statement: "This does not alter our adherence to PLOS ONE policies on sharing data and materials."

Thank you very much in behalf of our research group.

Respectfully,

Joan Dymphna P. Reaño, MD

Cardiology Fellow

Manila Doctors Hospital

---

## [Decision Letter · Decision Letter 1]

10 Feb 2020

A systematic review and meta-analysis on the effectiveness of an invasive strategy compared to a conservative approach in patients > 65 years old with non-ST elevation acute coronary syndrome

PONE-D-19-21413R1

Dear Dr. Reano,

We are pleased to inform you that your manuscript has been judged scientifically suitable for publication and will be formally accepted for publication once it complies with all outstanding technical requirements.

With kind regards,

Carmine Pizzi

Academic Editor

PLOS ONE

Reviewer #1: Yes: Fabrizio D'Ascenzo

---

## [Editor Report · Acceptance letter]

13 Feb 2020

PONE-D-19-21413R1 

A systematic review and meta-analysis on the effectiveness of an invasive strategy compared to a conservative approach in patients > 65 years old with non-ST elevation acute coronary syndrome 

Dear Dr. Reaño:

I am pleased to inform you that your manuscript has been deemed suitable for publication in PLOS ONE. Congratulations! Your manuscript is now with our production department. 

With kind regards,

on behalf of

Dr. Carmine Pizzi 

Academic Editor

PLOS ONE